

# Transcription factor NF-E2-related factor 2 plays a critical role in acute lung injury/acute respiratory distress syndrome (ALI/ARDS) by regulating ferroptosis

JiaLi Deng[1], Na Li[1], Liyuan Hao[1], Shenghao Li[1], Nie Aiyu[1], Junli Zhang[2] and XiaoYu Hu[3]

[1] School of Clinical Medicine, Chengdu University of Traditional Chinese Medicine, Chengdu, Sichuan, China
[2] Department of Infectious Disease, Jiangsu Province Hospital of Traditional Chinese Medicine, Nanjing, Jiangsu, China
[3] Department of Infectious Disease, Affiliated Hospital of Chengdu University of Traditional Chinese Medicine, Chengdu, Sichuan, China

Corresponding author
XiaoYu Hu, xiaoyuhu_dr@sina.com

## ABSTRACT

NRF2 is an important transcription factor that regulates redox homeostasis *in vivo* and exerts its anti-oxidative stress and anti-inflammatory response by binding to the ARE to activate and regulate the transcription of downstream protective protein genes, reducing the release of reactive oxygen species. Ferroptosis is a novel iron-dependent, lipid peroxidation-driven cell death mode, and recent studies have shown that ferroptosis is closely associated with acute lung injury/acute respiratory distress syndrome (ALI/ARDS). NRF2 is able to regulate ferroptosis through the regulation of the transcription of its target genes to ameliorate ALI/ARDS. Therefore, This article focuses on how NRF2 plays a role in ALI/ARDS by regulating ferroptosis. We further reviewed the literature and deeply analyzed the signaling pathways related to ferroptosis which were regulated by NRF2. Additionally, we sorted out the chemical molecules targeting NRF2 that are effective for ALI/ARDS. This review provides a relevant theoretical basis for further research on this theory and the prevention and treatment of ALI/ARDS. The intended audience is clinicians and researchers in the field of respiratory disease.

## INTRODUCTION

Acute lung injury (ALI) is defined as damage to alveolar epithelial cells and capillary endothelial cells due to a variety of intrapulmonary and extrapulmonary causative factors, leading to diffuse interstitial and alveolar edema and resulting in acute hypoxic respiratory insufficiency characterized by persistent hypoxemia and pulmonary infiltrates (*Mowery, Terzian & Nelson, 2020*; *Sweeney & McAuley, 2016*). Acute respiratory distress syndrome (ARDS) is a severe manifestation of ALI as described by the 2012 Berlin definition, which often occurs in the setting of pneumonia, sepsis, aspiration of gastric contents, or major

trauma (*Force et al., 2012*; *Matthay et al., 2019*). Epidemiology shows ALI/ARDS has a mortality rate of more than 30% and lacks effective treatment (*ARDS Definition Task Force et al., 2012*; *Meyer, Gattinoni & Calfee, 2021*).

Ferroptosis is a novel mode of iron-dependent, lipid peroxidation-driven cell death proposed by *Dixon Scott et al. (2012)* as a consequence of imbalance in cellular metabolism and redox homeostasis (*Jiang, Stockwell & Conrad, 2021*). Unlike apoptosis and autophagy, it is characterized by an increase in intracellular free iron, accumulation of lipid peroxides, intact nuclei, reduction of mitochondrial cristae, alteration of the bilayer density of mitochondrial membranes, and rupture of the outer mitochondrial membranes, which leads to peroxidation of the Golgi apparatus, endoplasmic reticulum, and lysosomes, among others, and ultimately leads to cell death (*Dixon Scott et al., 2012*). Studies have shown that ferroptosis is closely related to diseases such as tumors, ischemia/reperfusion (I/R) injury, and neurological disorders (*Mou et al., 2019*; *Ren et al., 2020*; *Stockwell et al., 2017*; *Yan et al., 2020*). In recent years, more and more studies have confirmed that ferroptosis plays an important role in the pathogenesis of lipopolysaccharide (LPS)-induced ALI, I/R-induced ALI, drowning lung injury, oleic acid (OA)-induced ALI, and radiation-induced lung injury (RILI) (*Li et al., 2020b*; *Qiu et al., 2020*; *Ye et al., 2020*; *Zhang et al., 2022*; *Zhou et al., 2019*).

The transcription factor nuclear factor erythroid 2-related factor 2 (NRF2) is a key regulator of the endogenous antioxidant response. In addition to the antioxidant response, NRF2 plays a critical role in various metabolic pathways such as iron/heme metabolism, protein homeostasis, carbohydrate and lipid metabolism, and apoptosis. In addition to antioxidant responses, NRF2 also plays a key role in various metabolic pathways such as iron/heme metabolism, carbohydrate metabolism, lipid metabolism, and apoptosis, and thus its function is critical for cell survival during increased oxidative or metabolic stress (*Dodson, Castro-Portuguez & Zhang, 2019a*; *Dodson et al., 2019b*; *Tebay et al., 2015*). The results of several studies have confirmed that NRF2 is able to alleviate oxidative stress, inhibit ferroptosis, and thus ameliorate ALI induced by multiple causes (*Song et al., 2023*; *Wang et al., 2023*; *Ye et al., 2020*). Gaining a deeper understanding of the mechanism of ferroptosis regulated by NRF2 may effectively improve the treatment of ALI/ARDS. Therefore, the aim of the review is to provide a relevant theoretical basis for further research on this theory and the prevention and treatment of ALI/ARDS. The audience is intended for clinical doctors and researchers in the fields of cell death, inflammation and lung injury.

## SURVEY METHODOLOGY

We searched related literature by pubmed using the key words (ferroptosis) AND (acute lung injury)/(NRF2) AND (acute lung injury)/(ferroptosis) AND (NRF2). However, the articles which were not research articles or reviews were excluded.

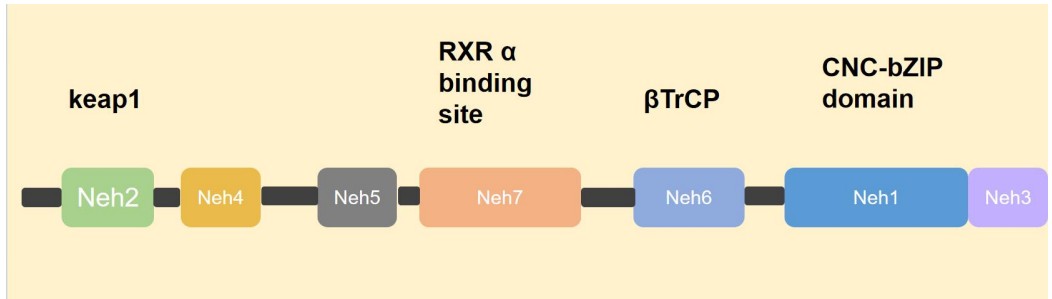

**Figure 1** **Structure of NRF2.** NRF2 consists of six Neh domains, namely Neh1–Neh6. Neh1 contains a CNC-bZIP structure that identifies and combines with ARE. Neh2 contains a motif that binds to the cytoplasmic protein Keap1. Neh3 domain, combined with chromatin to cleave CHD 6. The transcription process can only be initiated after Neh4, Neh5 and helper proteins are involved. Neh6 was β-TrCP identification. Neh7 domain, with RXRα Combining to prevent the recruitment of co-activators into the Neh4 and Neh5 domains, thereby inhibiting NRF2 activity.

## NRF2

### Structure of NRF2

NRF2 was first identified by *Moi et al. (1994)* while screening for proteins bound to the extended activator protein 1 (AP-1) sequence using a γgt 11 cDNA expression library. NRF2 is a 605-amino-acid CNC transcription factor that contains a cap'n'collar/basic leucine zipper (CNC-bZIP) structural domain and is therefore considered to be the third member of the mammalian CNC-bZIP protein family, named NRF2 (*Battino et al., 2018*; *Ma, 2013*). NRF2 contains six highly conserved NRF2-ECHhomology (Neh) structural domains, Neh1-Neh6 (*Baird & Dinkova-Kostova, 2011*; *Boutten et al., 2011*; *Ma, 2013*). Neh1 contains the CNC-bZIP structure, which is able to form a heterodimer with small Maf protein in the nucleus, thus recognizing and binding to the antioxidant response element (ARE) and initiating the transcription of target genes (*Itoh et al., 1997*). Neh2 is at the N-terminus and contains two highly conserved ETGE motifs and DLG motifs that bind to the cytoplasmic protein Keap1 (*Itoh et al., 1999*; *Katoh et al., 2005*; *Kobayashi et al., 2002*; *McMahon et al., 2003*; *McMahon et al., 2006*). Neh3 structural domain is at the C-terminus and binds the chromatin-deconjugating enzyme DNA-binding protein 6 (CHD 6) (*Nioi et al., 2005*). Neh4 and Neh5 are involved in initiating transcription of downstream genes. When NRF2-Maf enters the nucleus and binds to the ARE, it does not immediately initiate transcription but requires other auxiliary proteins to bind to Neh4 and Neh5 to initiate the transcription process (*Katoh et al., 2001*). Neh6 is a negative regulatory region that is not dependent on cellular redox status, and it contains two conserved peptide motifs, DSGIS and DSAPGS, which are recognized by β-TrCP (*Chowdhry et al., 2013*; *McMahon et al., 2004*). In 2013 researchers also identified the Neh7 structural domain, which binds to retinoid X receptor alpha (RXRα) and can inhibit NRF2 activity by preventing co-activator recruitment to the Neh4 and Neh5 structural domains (*Wang et al., 2013*) (Fig. 1).

## Target genes of NRF2 related to ferroptosis

The identified target genes of NRF2 are involved in various cellular processes such as redox regulation, lipid metabolism, iron homeostasis, detoxification of exogenous substances, transcriptional regulation, and DNA repair (*Dodson et al., 2019b*). Maintaining redox homeostasis and avoiding oxidative stress damage to nucleic acids, proteins, and lipid molecules are important functions of NRF2. Glutathione (GSH) is an important antioxidant in the body, and its synthesis is regulated by glutamate cysteine synthetase (γGCS) and glutathione synthetase (GSS). γ-GCS, also known as glutamate cysteine ligase (GCL), is the rate-limiting enzyme in the synthesis of GSH, and the activity of γ-GCS is affected by a variety of factors, including the expression of its catalytic subunit GCLM and GCLC, the degree of negative feedback inhibition of GSH, and functionally relevant post-translational modifications to specific sites on the GCL subunit. Glutathione peroxidase 4 (GPX4) utilizes GSH to reduce peroxides, and subsequently, glutathione reductase (GSR) reduces oxidized glutathione (GSSH), thereby recycling it. Thus, the target genes of NRF2 cover the entire process of GSH synthesis, antioxidant enzyme synthesis, and redox cycling. In addition, the system $Xc^-$ transporter protein (cystine-glutamate reverse transporter protein which also called SLC7A11, solute carrier family 7 member 11), which transports cysteine into the cell for biosynthesis, is also transcriptionally regulated by NRF2. Nicotinamide adenine dinucleotide phosphate (NADPH) is an important product of the pentose phosphate pathway as well as of the tricarboxylic acid cycle. Phosphate dehydrogenase (G6PD), phosphogluconate 6 dehydrogenase (PGD), transaldolase (TALDO 1), malic enzyme I (MEl), phosphoglycerate dehydrogenase (PHGDH) and isocitrate dehydrogenase I (IDHl) in both of these pathways are target genes of NRF2 (*Gorrini, Harris & Mak, 2013*; *Mitsuishi et al., 2012*; *Thimmulappa et al., 2002*). Finally, NRF2 controls intracellular iron homeostasis by regulating the transcription of heme oxygenase 1 (HO-1), an enzyme involved in heme catabolism, iron pump protein (FPN), and the light and heavy chains of ferritin (FTH 1/FLH 1) (*Agyeman et al., 2012*; *Alam et al., 1999*; *Harada et al., 2011*).

## Post translation regulation of NRF2

Regulation of NRF2 is largely controlled at the protein level.NRF2 is normally ligated to three E3 ubiquitin ligases and thus continuously targeted for degradation by the 26S proteasome: the CUL 3-RBX 1-Keap 1 complex, the SCF/β-TrCP complex, and the HMG-CoA reductase degradation 1 (HRD 1), which complexes mediate the degradation of NRF2 in specific regions and under different stimuli, respectively (*Harder et al., 2015*; *Rada et al., 2011*). Keap1 is a major regulator of NRF2. As a substrate bridging protein, Keap1 forms an E3 ubiquitin ligase complex with Cul 3 RING-box 1 (RBXl) (termed CRLKeap1) that is able to ubiquitinate NRF2 under normal conditions and anchors to actin, allowing NRF2 to be retained in the cytoplasm (*Sykiotis & Bohmann, 2010*). Upon stimulation by a pro-electronic or oxidative stressor, Keap1 spatial conformation is altered to fail to bind NRF2, and NRF2 is released from Keap1 and translocates into the nucleus, where it forms a heterodimer with small Maf proteins that binds to ARE sequences to turn on transcriptional regulation (*Boutten et al., 2011*; *Itoh et al., 1997*; *Niture et al., 2010*). NRF2 also controls Keap1 gene expression, and after the cell restores redox homeostasis, Keap1

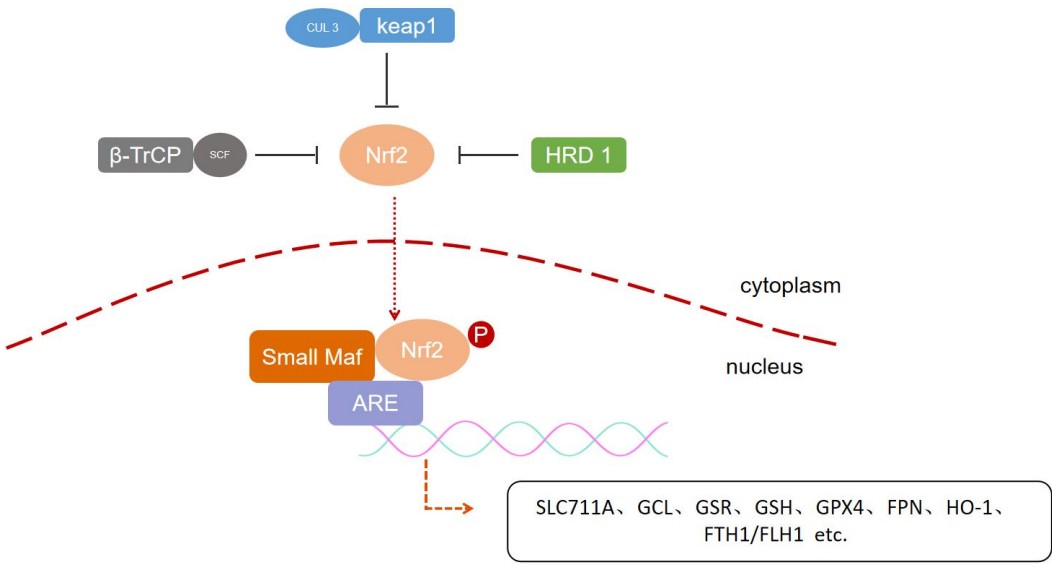

**Figure 2** **Post-translation regulation of NRF2.** Three types of E3 ubiquitin ligases include CUL 3-RBX 1-Keap 1 complex, the SCF/β-TrCP complex and HRD 1, respectively, mediate the degradation of NRF2 in specific regions and under different stimuli, regulating downstream gene expression.

translocates to the nucleus to bring NRF2 back into the cytoplasm, thereby repressing transcription of NRF2 target genes (*Sun et al., 2007*). Neh6 contains two motifs that can be recognized by β-TrCP, an F-box-containing protein whose C-terminus binds to the substrate through protein-protein interactions and whose F-box motif binds to the SCF E3 ubiquitin ligase complex, called SCFβ-TrCP (*Chowdhry et al., 2013*; *McMahon et al., 2004*; *Petroski & Deshaies, 2005*). ERAD-associated E3 ubiquitin-associated ligase synovial apoptosis inhibitor 1 (also called synoviolin or HRD1) is localized to the endoplasmic reticulum (ER), and during ER stress HRD1 interacts directly with NRF2, leading to ubiquitination and degradation of the CNC-bZIP transcription factor, thereby preventing NRF2 from coordinating the antioxidant response (*Wu et al., 2014*) (Fig. 2).

The NRF2-/- model is commonly utilized as a control to assess the involvement of relevant genes regulated by NRF2 in combating oxidative stress. However, recent studies have revealed that NRF2-/- cells display a distinct phenotype characterized by heightened activity of specific signal transduction pathways. Nevertheless, there are still gaps in our understanding of the tissue-specific effects of Nrf 2, which could aid in identifying the most appropriate disease targets for NRF2 activation therapy. NRF2 activation promotes cell restoration of REDOX and protein homeostasis, protects mitochondrial function, and inhibits inflammation, all processes that contribute to the survival of neurons and astrocytes, perhaps the most suitable disease areas are neurological diseases, as well as metabolic diseases and cancer prevention (*Dinkova-Kostova & Copple, 2023*; *Tebay et al., 2015*). In addition, the effects of NRF2 activation are complex, and the results are not always beneficial. NRF2 has a dual role in the development of cancer. In the early stage, NRF2 activation increases the expression of cell protection genes, generates antioxidant

mechanisms, and removes ROS, hydroxynonenal (HNE) and foreign organisms from cells, thereby restoring REDOX balance and avoiding unnecessary DNA mutations and cancer. However, in advanced cancer, up-regulation of NRF2 helps malignant cells tolerate high levels of ROS and avoid apoptosis by activating metabolic and cell protection genes that help enhance cell proliferation (*Kim & Keum, 2016*; *Menegon, Columbano & Giordano, 2016*). Furthermore, in some cases, the effects of NRF2 are complex and cell type-specific, such as in a mouse model of atherosclerosis, where loss of NRF2 in bone marrow-derived cells exacerbates atherogenesis, but loss of NRF2 overall increases plaque inflammation and vulnerability, thereby reducing lesion development (*Gao et al., 2013*; *Zhong et al., 2017*). Therefore, research on how to effectively control NRF2 activation and its degree of activation is necessary for the safe application of NRF2 activation therapy.

## LIPID PEROXIDATION AND FERROPTOSIS

### Lipid peroxidation

Lipid peroxidation is the process of oxidative degradation of lipids, and the chemical products of this oxidation are called lipid peroxides or lipid oxidation products. We can summarize this process as free radicals "stealing" electrons from lipids in cell membranes, leading to cellular damage, and as with any free radical reaction, the reaction involves three major steps: initiation, diffusion, and termination. Free radicals take bisallyl hydrogen atoms from polyunsaturated fatty acids (PUFAs) located in the carbon-carbon double bond to form phospholipid radicals (PL-), which are subsequently oxidized to phospholipid peroxyl radicals (PLOO-), which in turn take hydrogen atoms from another PUFA to produce PL- and phospholipid peroxide (PLOOH), which is then converted to PLOH by GPX4. If PLOOH is not converted to PLOH, PLOOH reacts with ferrous ions to form alkoxyphospholipid radicals (PLO-), which react with the PUFA to form PL- again, and so on in a cycle of non-stop proliferation that produces more PLOOH. when proliferation reaches the point where either two of the lipids, or the PLOO-, are present at a high enough concentration to interact with each other, or the endogenous The reaction stops when the antioxidant provides hydrogen atoms to form a stable non-radical product (*Conrad & Pratt, 2019*; *Yin, Xu & Porter, 2011*). The PUFAs of phospholipids in the lipid bilayer of cell membranes are particularly susceptible to lipid peroxidation because they contain multiple carbon-carbon double bonds (Fig. 3).

### Ferroptosis

As the name suggests, ferroptosis is an iron-dependent form of cell death. Increased free iron, accumulation of lipid peroxides, and a form of cell death morphologically distinct from apoptosis, necrotic cell death, and other forms of cell death characterize ferroptosis (*Dixon Scott et al., 2012*). Iron is one of the essential trace elements and the most abundant trace element in the human body. Iron ions usually take the form of trivalent iron and bind to transferrin, which binds to the transferrin receptor on the cell surface and transports the trivalent iron into the cell. Subsequently, the trivalent iron is reduced to divalent iron by the metal reductase STEAP3 to form a variety of iron complexes, which are involved in a variety of subsequent physiological and biochemical

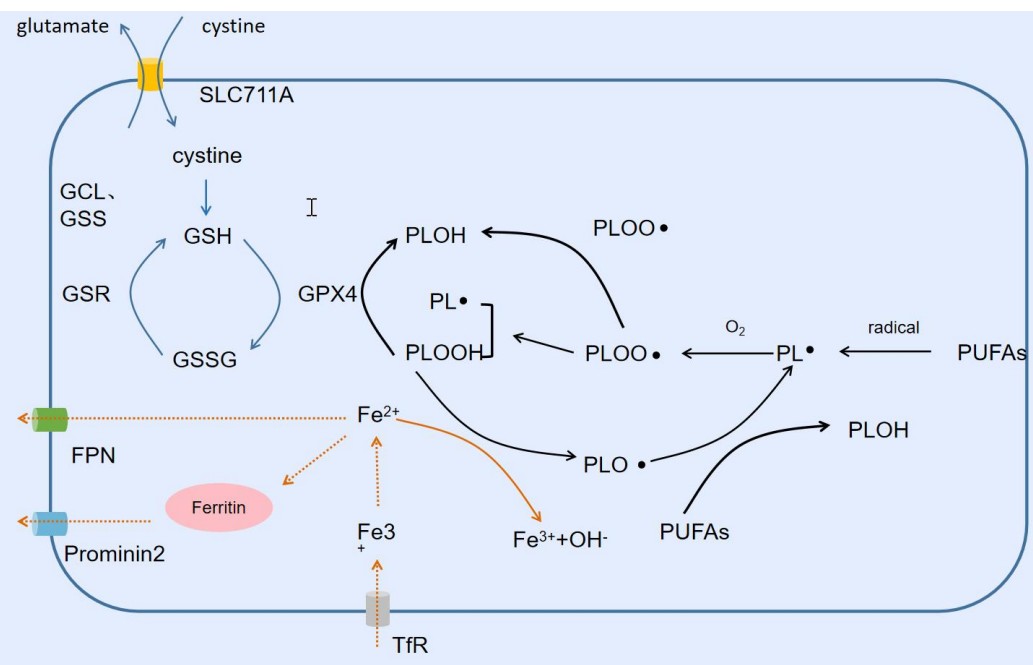

**Figure 3 Lipid peroxidation and ferroptosis.** Free radicals trigger the Fenton reaction to produce lipid peroxides, and SLC711A takes up cystine to synthesize GSH, which is used by GPX4 to reduce lipid peroxides. $Fe^{2+}$ is exported from FPN extracellularly or converted into ferritin Prominin2 for output.

processes. When the content of these complexes approaches saturation, excess divalent iron accumulates in the cell, forming an unstable iron pool. As mentioned earlier, the oxidation of ferrous iron in the unstable iron pool to trivalent iron is accompanied by the reduction of PLOOH to PLO-, which reacts with PUFAs to generate PL- again. This cycle continually generates more PLOOH and lipid peroxides, and the accumulation of PLOOH can rapidly lead to lethal membrane damage, thereby inducing cell death. Thus free iron is one of the key factors triggering ferroptosis (*Kakhlon & Cabantchik, 2002*). The alteration of intracellular iron pool capacity is inextricably linked to the export or import of iron ions and the synthesis and degradation of iron-containing proteins (*Anderson & Frazer, 2017*; *Ponka, 1999*). Iron regulatory protein 1/2 (IRP1/2) reduces the iron pool capacity by promoting the synthesis of ferritin, heme, and iron-sulfur proteins; FPN and ferritin transfer protein (Prominin2) transport iron ions and ferritin to the outside of the cell, respectively (*Brown et al., 2019*; *Pantopoulos, 2004*). In contrast, ferric ions enter the cell *via* the transferrin/transferrin receptor (TF/TFR-1) transport system or the SLC39A14 channel. Nuclear receptor coactivator 4 (NCOA4) and HO-1 respectively degrade ferritin and heme, to produce ferrous ions and thus increase the iron pool capacity (*Kawabata, 2019*; *Santana-Codina, Gikandi & Mancias, 2021*). Lipid peroxide scavenging involves two important systems $X_c^-$ and GPX4. $X_c^-$ takes up cystine from outside the cell to provide raw material for the synthesis of GSH. GPX4 is a key enzyme in the scavenging of PLOOH. GPX4 utilizes its catalytic selenocysteine residue and two electrons supplied by GSH to reduce

PLOOH to its corresponding alcohols to fulfill its detoxification. Not only that, GSSH can be reduced to GSH by GSR and thus recycled. Morphologically, ferroptosis is different from the regulation of cell death, necrotic cell death, and pyroptosis in that the nucleus is intact and there is no shrinkage or rupture of the nucleus (*Bertheloot, Latz & Franklin, 2021*; *Brigelius-Flohé & Maiorino, 2013*; *D'Arcy, 2019*; *Ketelut-Carneiro & Fitzgerald, 2022*). The morphological changes were mainly focused on mitochondrial alterations, such as mitochondrial crumpling, reduced mitochondrial cristae, increased mitochondrial density, and rupture of the outer mitochondrial membrane. In addition to this, the biochemical features of ferroptosis which are also markedly different from those of other cell death modalities include free iron increasing, accumulation of lipid peroxidation products, GSH depletion, and inhibition of the System $X_c^-$ and GPX4 (*Kazan & Kalaipandian, 2019*).

## Ferroptosis in ALI/ARDS

ARDS is a syndrome of acute respiratory failure caused by noncardiogenic pulmonary edema and is a severe manifestation of ALI, usually caused by pneumonia, sepsis, aspiration of gastric contents, or major trauma, but also by pancreatitis and drug reactions (*Matthay et al., 2019*). The diagnostic criteria for ARDS, as defined by the 1994 American-European consensus definition (AECC), include an acute episode of impaired oxygenation (arterial hypoxemia, $PaO_2/FiO_2$ ratio <300 mmHg) and a bilateral infiltrate on chest imaging, whereas left atrial hypertension is not recognized as a major cause of pulmonary edema. Cause (*Bernard et al., 1994*). The 2012 Berlin definition classifies ARDS into three categories according to the degree of hypoxemia: mild (PaO2/FiO2 200–300 mmHg), moderate (PaO2/FiO2 100–200 mmHg), and severe PaO2/FiO2 <100 mmHg) (*Force et al., 2012*). ALI is characterized by diffuse pulmonary infiltrates, persistent hypoxemia, and respiratory distress; its pathological changes include damage to pulmonary capillary endothelial cells and alveolar epithelial cells, and diffuse alveolar and interstitial edema. The pathogenesis of ALI is very complex, and it begins with the activation of the innate immune system by endogenous molecules associated with microbial products or cellular injury to generate reactive oxygen species, chemokines, cytokines, *etc*. A large number of reactive oxygen species and cytokines are also generated. cytokines, *etc*. Large amounts of reactive oxygen species (ROS) and inflammatory factors trigger an imbalance between the oxidative and antioxidant systems, exacerbating alveolar epithelial and endothelial cell damage while fighting infection. In turn, the damage leads to increased endothelial and epithelial permeability and alveolar fluid accumulation, with cell necrosis and fluid accumulation in turn triggering a more severe inflammatory and immune response, creating a vicious cycle (*Opitz et al., 2010*; *Ornatowski et al., 2020*; *Zheng et al., 2022*). Despite decades of research, there is still no effective therapeutic drug for ALI, and its clinical treatment is still based on respiratory support and fluid management (*Matthay et al., 2019*). In recent years, a large number of studies have shown that ferroptosis is related to the occurrence and development of ALI and is a potential target for ALI treatment. LPS can induce ferroptosis in lung cells both *in vivo* and *in vitro*. Ferrostatin-1 (Fer-1), an ferroptosis inhibitor, reduces the effects of pro-inflammatory cytokines interleukin (IL)-6 and tumor necrosis factor (TNF-α) in bronchoalveolar lavage fluid have a therapeutic effect on LPS-induced ALI (*Liu*

*et al., 2020*). HO-1 is a double-edged sword in ferroptosis; on the one hand, its promotion of heme catabolism leads to an increase in free iron and promotes the occurrence of ferroptosis, and on the other hand, HO-1 is an antioxidant enzyme against oxidative stress and cell death (*Loboda et al., 2016*). Isoliquiritin apioside attenuates intestinal I/R-induced lung injury and ferroptosis by inhibiting hypoxia inducible factor-1α (HIF-1α)/HO-1 under hypoxic conditions (*Zhongyin et al., 2022*). It was also found that the lung tissues of mice with OA-induced ALI showed typical morphological changes of iron apoptosis, GSH depletion and GPX4 down-regulation, lipid peroxidation, and iron accumulation, suggesting that iron apoptosis may be involved in the pathogenesis of ALI (*Zhou et al., 2019*). These features related to ferroptosis found in lung tissues suggest that ferroptosis is one of the pathological mechanisms in the development and progression of ALI, and therefore targeting ferroptosis may be an effective treatment for ALI.

## Biology and role of ferroptosis in disease

Detailed analysis of p53's specific lysine acetylation sites showed that p53 can enhance ferroptosis by inhibiting the transcription of SLC7A11, which may contribute to p53's tumor suppressor function *in vitro* and *in vivo* (*Jiang et al., 2015*). p53 also activates the expression of the gene lipoxygenase (ALOX)12 transcribed in the 17p region of chromosome that is normally lost in tumors (*Chu et al., 2019*). Similar to p53, tumor suppressor and epigenetic regulator ubiquitin carboxyl-terminal hydrolase (BAP1) also promote ferroptosis by down-regulating SLC7A11 expression (*Zhang et al., 2018*). Knockout of MLL4 (histone-lysine N-methyltransferase 2B) drives increased expression of SLC7A11, GPX4, and stearoyl-coenzyme A desaturase 1 (SCD1) in mice, as well as loss of expression of lipoxygenases ALOX 12, ALOX 12B, and ALOX 3, which increases cell resistance to ferroptosis and ultimately leads to the development of precancerous skin lesions (*Egolf et al., 2021*). Other research has demonstrated that CD8+ T cells exert a tumor suppressor function by inducing ferroptosis in tumor cells. Interferon gamma (IFN-γ) secreted by activated CD 8+ cells down-regulates SLC7A11 leading to ferroptosis in tumor cells (*Wang et al., 2019*). IFN-γ also upregulates acyl-CoA synthetase long-chain family member 4 (ACSL4), which contributes to the incorporation of long-chain PUFA into the plasma membrane and increases cell sensitivity to ferroptosis (*Liao et al., 2022*). In addition, aging in *C. elegans* and cell death in *Magnaporthe Oryzae* were also associated with ferroptosis (*Jenkins et al., 2020*; *Shen et al., 2020*).

In addition to the above physiological functions, ferroptosis also involves a variety of pathologies. For example, ferroptosis inhibitors improved non-septic multiple organ damage induced by iron sulfate in mice, reducing the expression of injury markers in the kidneys, liver, muscle, heart, and plasma (*Van Coillie et al., 2022*). Under the control of fatty acid desaturase 2 (FADS2), iron death activation in host cells inhibits hepatitis C virus replication.

Knocking out FADS2 using RNA interference enhances HCV replication, so, in the case of HCV, iron death helps suppress viral replication (*Yamane et al., 2022*). Activation of ferroptosis was also detected in autoimmune disease models. Li et al. found that neutrophil counts were low in Systemic lupus erythematosus (SLE) patients, and serum of SLE patients

could induce neutrophil death, which might be caused by GPX4 down-regulation mediated by CaMKIV-CREMα. Furthermore, mice with neutrophil-specific GPX4 knockout also had SLE characteristics (*Li et al., 2021b*). Along with organ injury, infectious diseases, and autoimmune diseases, ferroptosis has been linked to several neurodegenerative diseases, including Alzheimer's disease, Parkinson's disease, and Huntington disease (*Stockwell et al., 2017*). Another report suggests that Prussian blue nanoparticles inhibited the degeneration and death of retinal pigment epithelium cells in retinal diseases by reducing the availability of ferrous iron, suggesting that retinal diseases may also be involved in iron death activation (*Tang et al., 2021*).

## NRF2 INVOLVED IN ALI/ARDS

NRF2 is a key transcription factor that protects the body from oxidative stress and regulates inflammatory responses, and as mentioned above, its role in inhibiting lipid peroxidation and ameliorating ferroptosis is clear. Many studies targeting NRF2 have demonstrated that ferroptosis can be inhibited by modulating the NRF2 signaling pathway, thereby ameliorating ALI/ARDS induced by multiple causes.

### NRF2 in LPS induced ALI/ARDS

Sepsis is a dysregulation of the body's immune response to infection, resulting in life-threatening tissue damage and organ dysfunction. It is characterized by a dysregulated systemic inflammatory response as well as immunosuppression, with rapid progression, poor prognosis, and a mortality rate of more than 20% (*Rudd et al., 2020*; *Singer et al., 2016*). Septic ALI is actually an acute lung tissue injury caused by a dysregulated inflammatory response. Previous studies have demonstrated the ability to counteract LPS-induced oxidative stress, attenuate the inflammatory response, inhibit autophagy, apoptosis, as well as pyroptosis, and ameliorate lung injury by activating the NRF2 signaling pathway (*Dhlamini et al., 2022*; *Huang et al., 2020*; *Kong et al., 2021*; *Liu et al., 2021*). With more and more studies showing that ferroptosis occurs in LPS-induced ALI/ARDS, the role of NRF2 in LPS-induced ferroptosis in ALI/ARDS has attracted extensive research. It was found that LPS treatment significantly decreased the levels of SLC 7A 11 and GPX 4, while malondialdehyde (MDA) and total iron levels were significantly increased, and Fer-1 reversed this result, suggesting that LPS treatment induced the occurrence of ferroptosis in ALI (*Liu et al., 2020*). By promoting the expression of NRF2 or knocking down its negative regulators could reduce the secretion of various cytokines such as TNF-α, IL-1b and IL-6 as well as the production of ROS and MDA, decrease the level of ferrous ions, and increase the expression of SLC 7A 11 and GPX 4 to ameliorate the LPS-induced ferroptosis in ALI, suggesting that the NRF2 signaling pathway plays a critical role in improving the pathogenesis of LPS-induced ferroptosis in ALI (*Nishizawa, Yamanaka & Igarashi, 2022*; *Wang et al., 2023*).

### NRF2 in Intestinal I/R-Induced ALI/ARDS

Intestinal I/R (IIR) often occurs after intestinal obstruction, small bowel torsion, acute mesenteric ischemia, shock, and severe trauma, usually cause damage to the intestinal
mucosal barrier and intestinal toxins or bacterial translocation which lead to systemic inflammation, tissues and organs damage, and even multi-organ dysfunction (*De Perrot et al., 2003*; *Meng et al., 2017*; *Mura et al., 2007*). Recent studies have shown that IIR-induced ALI/ARDS is increasingly associated with ferroptosis. *In vitro* and *in vivo* studies found that ferroptosis occurred in alveolar epithelial cells in IIR-induced ALI/ARDS (*Li et al., 2020b*; *Zhongyin et al., 2022*). It was found that regulating the levels of its downstream ferroptosis-related proteins by NRF2 could counteract oxidative stress and attenuate IIR-induced ferroptosis in ALI/ARDS, *e.g.*, NRF2 regulated the levels of SLC7A11, which increased GSH synthesis and enhance the reduction of lipid peroxides by GPX4 (*Dong et al., 2020*; *Dong et al., 2021*; *Qiang et al., 2020*). In addition, *in vitro* administration of isoliquiritin apioside (IA) inhibited IIR-induced up-regulation of Hif-1α mRNA as well as protein in ALI/ARDS mice or MEL12 cells and reduced the accumulation of lipid peroxidation products and $Fe^{2+}$ in IIR-induced lung tissues, suggesting that IA inhibit ferroptosis through the HIF-1α/HO-1 pathway during ALI (*Zhongyin et al., 2022*). However, *Li et al. (2020b)* found that the protective effect of iASPP overexpression on IIR-induced ferroptosis in ALI/ARDS was dependent on the expression of NRF2, which increased the expression of GPX4 as well as HIF-1α, and decreased the expression of ferroptosis-associated proteins FTH1, NQO-1, and HO-1 (*Li et al., 2020b*). Hypoxia inducible factor (HIF) is a heterodimer consisting of a constitutive β-subunit and oxygen-dependent α-subunits (HIF-1α and HIF-2α), and the HIF-1β subunit is stably expressed in the cytosol, whereas the HIF-1α subunit is degraded by the ubiquitin-monoprotease hydrolysis complex right after translation. In hypoxia conditions, the degradation of the HIF-1α subunit is inhibited, and the α and β subunits form active HIF-1 which is transferred to the nucleus to regulate the transcription of various genes (*Ke & Costa, 2006*; *Lee et al., 2004*). The target genes include Lactate dehydrogenase A (LDHA), phosphofructokinase L (PFKL), phosphoglycerate kinase 1 (PGK1), hexokinase, HO-1 and other energy metabolism and iron metabolism related genes (*Ke & Costa, 2006*; *Ma, 2013*). Both of the above studies demonstrated that inhibition of HO-1 is beneficial for the protection of ferroptosis in ALI/ARDS, but the relationship between HIF-1α and NRF2 in ferroptosis in ALI/ARDS, and whether they interact with each other and regulate ferroptosis remain to be investigated in depth.

## NRF2 in radiation-induced ALI/ARDS

Radiation therapy is one of the most important therapeutic modalities in the treatment of malignant tumors. Radiation therapy for malignant tumors and radiation accidents can lead to radiation damage. The main causes are radiation-induced DNA damage and radiation-generated reactive oxygen species leading to lipid peroxidation-induced cell death (*Jiao, Cao & Liu, 2022*; *Morgan & Lawrence, 2015*). Radiation-induced ferroptosis in ALI (RILI) was inhibited by the use of Fer-1, an ferroptosis inhibitor, which activates the NRF2 signaling pathway, increases antioxidant proteins, and decreases ROS production (*Li et al., 2022b*). The researchers found a significant decrease in GPX4 levels and an increase in ROS content in the RILI mouse model cells, which were reversed by treatment with the ferroptosis inhibitor Liproxstatin-1 (*Li, Zhuang & Qiao, 2019b*). Moreover, Liproxstatin-1 activated the NRF2 pathway in RILF (radiation-induced lung fibrosis), which up-regulated

HO-1 and NQO1 levels, down-regulated the level of transforming growth factor (TGF)-β1, and attenuated ROS and inflammatory injury (*Li et al., 2019a*).

## NRF2 in drowning and oleic acid-induced ALI/ARDS

Drowning is one of the three leading causes of accidental death, resulting in more than approximately 360,000 deaths per year worldwide (*Abelairas-Gomez et al., 2019*; *Handley, 2014*; *The Lancet, 2017*). ALI is one of the most common complications of drowning, which damages alveolar epithelial cells, leading to hypoxia, hemorrhage, and oxidative stress as well as inflammation, ultimately develop into ARDS (*Jin & Li, 2017*; *Li et al., 2018*; *Liu et al., 2014*). Recent studies have shown that oxidative stress plays an important role in the pathogenesis of drowning-induced ALI. Studies have shown that activation of NRF2 increases cell viability, decreases intracellular ROS and lipid ROS levels, prevents glutathione depletion and lipid peroxide accumulation, and improves mitochondrial function (*Qiu et al., 2020*). OA-induced lung injury causes morphological and cellular changes similar to those observed in human ALI and ARDS and is a widely used laboratory model of ALI. In OA-induced ALI, the direct toxicity of OA to the pulmonary vascular endothelium leads to an increase in pulmonary vascular permeability and the extravasation of fluid in the pulmonary vasculature (*Motohiro et al., 1986*; *Schuster, 1994*; *Syrbu, Thrall & Smilowitz, 1996*; *Wang, Bodenstein & Markstaller, 2008*). The investigators found that OA-induced ALI lung tissue iron concentration was significantly increased, ferritin was decreased, GSH and GPX 4 proteins were decreased, mitochondrial atrophy and mitochondrial membrane rupture were observed in lung cells, as well as an increase in mRNA expression of prostaglandin endoperoxide synthase 2 (PTGS2), which was 7-fold that of the control group (*Zhou et al., 2019*). sulforaphane up-regulated NRF2 levels in serum and lung tissues of the OA-induced ALI model and inhibited ARDS (*Sun et al., 2018*). It suggests that ferroptosis does occur in the OA-induced ALI model and that the prognosis of the OA-induced ALI model can be improved by activation of NRF2, but more specific modulation needs to be further investigated in drowning and OA-induced ALI ferroptosis, as well as the role of NRF2 by modulating which molecules improve its outcome, which still needs to be investigated further and at a deeper level.

## Potential challenges in targeting NRF2 as a therapeutic approach

The target specificity of NRF2 activators is crucial for the treatment of related diseases. Some NRF2 activators are electrophilic molecules targeting cysteine 151 in KEAP1 (*Dayalan Naidu et al., 2018*; *Zhang & Hannink, 2003*), leading to KEAP1 alkylation that prevents it from binding to NRF2, thereby promoting its accumulation and downstream gene transcription. Therefore, such activators not only affect KEAP1 but also other proteins containing cysteine (*Sauerland & Davies, 2022*; *Zaro et al., 2019*). In fact, the anti-inflammatory effects of many electrophilic NRF2 activators are partially NRF2 independent (*Ryan et al., 2022*), such as inhibition of innate immune kinase interleukin-1 receptor associated kinase 4 (IRAK4) (*Zaro et al., 2019*). In recent years, researchers have developed KEAP 1-NRF2 protein-protein interaction (PPI) inhibitors targeting the Kelch domain of KEAP 1. However, NRF2 is not the only binding partner of KEAP1, and several proteins
(such as Bcl2, IKKβ, *etc.*) have been shown to bind to KEAP1's Kelch domain (*Hast et al., 2013*). In a clinical trial, it was observed that patients with type 2 diabetes and stage 4 chronic kidney disease had elevated levels of serum liver injury markers, such as alanine aminotransferase (ALT), aspartate aminotransferase (AST), and γ-glutamyl transferase, after receiving methylprednisolone intervention for 4 weeks (*Lewis et al., 2021*). The subsequent studies of these findings in cells and animal models revealed that the increase in ALT was attributed to NRF2 activation. However, the modest upregulation of related gene expression could not fully account for the up to 20-fold increase in serum ALT/AST detected in a few patients after drug therapy (*Church & Watkins, 2021*). So is there a risk of hepatotoxicity associated with NRF2 activation? The frequency of administration of NRF2 activators should also be taken into consideration. Studies using mouse models have shown that after a single dose of an NRF2 activator is administered, the duration (days) of relevant downstream gene expression exceeds the half-life (hours) of the drug (*Knatko et al., 2015*). Long-term treatment with an NRF2 activator three times a week was sufficient to improve inflammatory response and pathological damage in animal models of nonalcoholic fatty liver disease and liver fibrosis (*Sharma et al., 2018*). Therefore, there are still many problems to be solved before NRF2 can be used as a target for clinical treatment.

## THE PROTECTIVE EFFECT OF COMPOUND MOLECULES ON ALI MODELS BY REGULATING THE NRF2 SIGNALING PATHWAY AND FERROPTOSIS

NRF2 has been widely studied in a variety of diseases, and its role in combating oxidative stress and reducing inflammation has been confirmed. Not only that, activation of the NRF2 signaling pathway can effectively inhibit ferroptosis and reduce tissue damage. Many natural drugs, chemicals, and even hormones can improve ferroptosis by activating NRF2 signaling pathway. For example, Icariin and Astragaloside IV activate NRF2/HO-1 pathway which attenuates redox imbalance to reduce ferroptosis in endplate chondrocytes and brain cells, similarly, there are many alike studies in ALI (*Liu et al., 2022b*; *Shao et al., 2022*). Obacunone, Uridine, and Itaconate can block the degradation of NRF2 and promote the transcription of target genes, including GPX 4, GCLM, SLC7A11, and HO-1, thereby inhibiting cellular ferroptosis and attenuating LPS-induced ALI/ARDS (*He et al., 2022*; *Lai et al., 2023*; *Li et al., 2022a*).

Experimental results have shown that the half-life of NRF2 protein is 16.9 minutes in the absence of Obacunone, but in the presence of Obacunone, the half-life of NRF2 protein increases to 35.9 minutes, indicating that Obacunone stabilizes NRF2 protein. When we treated cells with Obacunone and the known NRF2 activator SF separately, we observed that the ubiquitination inhibition of NRF2 by Obacunone was similar to that of the positive control SF (*Xu et al., 2016*). These data support that Obacunone can activate NRF2 dependent responses by increasing the stability of NRF2 while inhibiting its ubiquitination. In human HEK 293 T cells, overexpression of Myc-DDK labeled KEAP 1 was observed, and the cells were treated with OI (Itaconate derivative). Immuno-precipitation tandem mass spectrometry was used to detect the presence of KEAP 1 peptide containing Cys 151.

**Table 1  Compound molecules regulate ferroptosis to reduce ALI with NRF2.**

| Name | Source | Target | Model | Ref |
|---|---|---|---|---|
| Obacunone | Citrus fruits | Nrf2/HO-1 | LPS-C57BL/6 mice | *Li et al. (2022a)* |
| Panaxydol | *Panax ginseng* | Keap1-Nrf2/HO-1 | LPS-C57BL/6 mice | *Li et al. (2021a)* |
| UrolithinA (UA) | Ellagitannins | Keap1-Nrf2/HO-1 | LPS-C57BL/6 mice | *Lou et al. (2023)* |
| Ferulic acid | Plants and vegetables | Nrf2/HO-1 | CLP-BALB/c mice | *Tang et al. (2022)* |
| Astaxanthin | Sea food | Keap1-Nrf2/HO-1 | LPS- mice | *Luo et al. (2022b)* |
| Itaconate | Cisaconitate | Keap1-Nrf2 | LPS-C57BL/6 mice | *He et al. (2022)* |
| Uridine | Metabolites | Nrf2/GPX4/HO-1 | LPS-C57BL/6 mice | *Lai et al. (2023)* |
| Mucin 1 | Metabolites | GSK3 β/Keap1-Nrf2/GPX4 | CLP-C57BL/6 mice | *Wang et al. (2022)* |

OI treatment increased the mass of the peptide by 242.15 Da, consistent with OI alkylation. Alkylation of KEAP 1 promoted NRF2 accumulation and increased the expression of downstream genes with antioxidant and anti-inflammatory abilities (*Mills et al., 2018*). In addition, astaxanthin, panaxydol, urolithin(A), and ferulic acid could effectively attenuate LPS-induced inflammation and ferroptosis, reduce the degree of pulmonary edema and inflammatory cell infiltration by regulating the Keap 1-NRF2-HO-1 pathway to increase the expression of GPX4 and SLC7A11 and decrease the level of $Fe^{2+}$, and significantly reduce the degree of lung histopathology (*Li et al., 2021a*; *Lou et al., 2023*; *Luo et al., 2022b*; *Tang et al., 2022*). However, when cells were transfected by NRF2 siRNA or shRNA, the anti-ferroptosis and anti-inflammatory effects of astaxanthin and ferulic acid were abolished (*Luo et al., 2022b*; *Tang et al., 2022*). Similar to the aforementioned results, if BEAS-2B cells were treated with 40 μg/ml Panaxydol, SnPP (HO-1 inhibitor) and ML 385 (Nrf-2 inhibitor) for 24 hours, Panaxydol's anti-ferroptsis and anti-inflammatory effects were significantly diminished (*Li et al., 2021a*). Keap 1-NRF2-GPX4 is also an important pathway to regulate ferroptosis which could be regulated by Mucin 1. Mucin 1 reduces $Fe^{2+}$ and MDA, increases GSH and GPX4 expression, and improves sepsis-induced ALI through the Keap 1-NRF2-GPX4 pathway. By using MUC 1 inhibitors, it is observed that Keap 1 expression is elevated, GSK 3β phosphorylation level is decreased, and NRF2 cannot enter the nucleus, thereby reducing the expression of GPX 4 (*Wang et al., 2022*). This shows that the NRF2 signaling pathway plays an important role in protecting lung tissues from acute injury due to various causes and is one of the keys to studying ALI/ARDS. Although we have gained some understanding of the pathways of these compounds through the aforementioned literature, further research is needed on their specific mechanisms of action (Table 1).

## DISCUSSION

ALI/ARDS is usually caused by pneumonia, sepsis, trauma or drugs. Through activation of the innate immune system, ROS, chemokines and cytokines are produced, *etc.* The large amount of ROS and inflammatory factors triggers an imbalance between the oxidative and antioxidant systems, leading to pathological changes such as injury to pulmonary capillary endothelial cells and alveolar epithelial cells, and diffuse alveolar and interstitial edema.

Several studies have shown that ferroptosis plays an important role in acute lung injury. In mouse models of ALI/ARDS, ferroptosis activators aggravated alveolar inflammation and pulmonary edema, increasing inflammatory factor levels, and these effects were reversed by ferroptosis inhibitors (*Dong et al., 2020*; *Liu et al., 2020*). As an important regulator of redox homeostasis, NRF2 plays an important protective role in ALI/ARDS. Characteristic manifestations of ferroptosis such as GSH depletion and downregulation of GPX4, lipid peroxidation and iron accumulation have been found in ALI induced by multiple causes (LPS, I/R, OA *etc.*). The target genes of NRF2 include more than 500 genes involved in oxidative stress (HO-1, GPX4), iron metabolism (FPN, FTH1/FTL1), as well as apoptosis (Bcl-2, B-cell lymphoma-2), autophagy (p62) and other genes (*Audousset, McGovern & Martin, 2021*; *de la Vega et al., 2016*). Such as GPX4, one of the proteins transcribed by NRF2. Inhibition of GPX4 leads to accumulation of lipid peroxide and ferroptosis (*Li et al., 2020a*). NRF2 can directly or indirectly up-regulate the expression of GPX4 and protect cells from ferroptosis. Inhibition of Keap1 expression with Keap1 siRNA significantly promoted the accumulation of NRF2 and increased the expression level of GPX 4, and finally alleviated lung tissue damage in sepsis. In contrast, the expressions of GPX 4, FTH1 and HO-1 in the lung tissues of Nrf2-/- mice were significantly reduced, and ferroptosis and oxidative stress were significantly higher than WT mice (*Li et al., 2020c*). Are these proteins regulated by NRF2 that help control ferroptosis acting alone or in combination? Is there a core protein that is the key mechanism by which NRF2 regulates ferroptosis? Future studies are needed to confirm these questions.

Except ferroptosis, NRF2 has been extensively studied in autophagy and pyroptosis. Autophagy is considered a self-protective mechanism of the organism, which is a normal dynamic life process in which cells utilize lysosomes to degrade and selectively remove their damaged, aged or excess biomolecules and organelles, releasing free small molecules for cellular recycling (*Filomeni, De Zio & Cecconi, 2015*; *Glick, Barth & Macleod, 2010*; *Racanelli et al., 2018*). It was found that NRF2 knockdown upregulated autophagosomes and autophagy-related proteins in LPS-interacted BEAS-2B cells (*Kong et al., 2021*). However, in LPS-interacted macrophages, NRF2 deficiency inhibited autophagy and promoted M1 macrophage polarization and inflammation (*Luo et al., 2022a*). Overexpression of NRF2 improved macrophage autophagy, inhibited phosphorylation of p65 in the nucleus and thus inhibited nuclear factor-κB (NF-kB) expression, and ultimately regulated macrophage transformation from M1 to M2 type and reduced tissue inflammation (*Huang et al., 2019*; *Luo et al., 2022a*; *Wei et al., 2018*). The different expression of NRF2 on autophagy in epithelial cells and macrophages may be related to the cell type, which still needs further study. Pyroptosis, also known as cellular inflammatory necrosis, occurs when inflammatory vesicles activate part of the caspase family of proteins to cleave and activate gasdermin proteins, and activated gasdermin proteins translocate to the membrane, forming pores and causing a loss of cellular integrity, leading to the release of cellular contents, increased permeability, and the triggering of inflammatory responses that ultimately lead to rupture and lysis of the cell membrane (*Feng et al., 2022*; *Wei et al., 2022*; *Yu et al., 2021*). It was found that inhibition of p65 phosphorylation by NRF2 not only inhibits the inflammatory pathway but also prevents the synthesis of NOD-like

receptor protein 3 (NLRP3) inflammatory vesicles and reduces the occurrence of cellular pyroptosis (*Liu et al., 2022a*; *Xue et al., 2022*).

In addition, extensive studies of the NRF2 signaling pathway have shown that several kinases target NRF2 expression, including phosphoinositide 3-kinase/Akt (PI3K/Akt), 5′-AMP-activated protein kinase (AMPK), and mitogen-activated protein kinase (MAPK). PI3K/Akt can reduce NRF2 degradation by inhibiting glycogen synthase kinase 3β (GSK-3β) or promoting the expression of the mammalian target of rapamycin 1 (mTORC 1). Similar to Keap1, GSK-3β activates Fyn, which results in tyrosine phosphorylation of NRF2 protein then leading to export of NRF2 to the nucleus and its degradation, inhibiting the expression of related genes. By activating the PI3K/Akt/GSK-3β/NRF2 intracellular signaling pathway, NRF2 activation and up-regulation of NRF2 expression can prevent cardiomyopathy and can also increase the expression of HO-1 to play a neuroprotective role (*Hui et al., 2017*; *Xin et al., 2018*). mTORC 1 phosphorylates p62 to compete with NRF2 for binding to Keap1, and this interaction allows p62 to chelate Keap1 into autophagosomes, thus preventing Keap1-mediated degradation of NRF2 and leading to activation of the NRF2 pathway (*Ichimura et al., 2013*). Studies have confirmed that Acetovanillone can activate the PI3K/Akt/mTOR/NRF2 pathway thereby preventing cyclophosphamide-induced acute lung injury (*Abd El-Ghafar et al., 2021*). However, another study showed that curcumin inhibited PI3K/Akt/mTOR and enhanced the NRF2/HO-1 signaling pathway, thereby suppressing renal oxidative stress and improving renal function in Heymann nephritis rats (*Di Tu et al., 2020*). PI3K/Akt/mTOR inhibition usually down-regulates NRF2 expression, but this study showed enhancement of the NRF2/HO-1 signaling pathway, could it possibly be related to the activation of AMPK signaling? AMPK, as an important kinase regulating energy homeostasis, is one of the central regulators of eukaryotic cell and organism metabolism, responsible for supervising the cellular capacity inputs and outputs and maintaining a smooth cell physiological activity operation (*Herzig & Shaw, 2018*; *Steinberg & Hardie, 2023*). Like PI3K/Akt, AMPK promotes NRF2 expression by inhibiting GSK-3β too. AMPK inhibitors block p-AMPK, p-GSK-3β, and NRF2 protein expression which reversed the protective effect of xanthohumol against LPS-induced ALI (*Lv et al., 2017*). In addition, three kinases in the MAPK pathway, extracellular signal-regulated kinase1/2 (ERK1/2), c-Jun N-terminal kinase (JNK), and p38MAPK, were able to positively regulate NRF2 (*Matzinger, Fischhuber & Heiss, 2018*). Different substances can activate NRF2 and promote its nuclear translocation by activating different MAPK kinases to regulate downstream gene expression. For example, cinnamaldehyde activates ERK 1/2, Akt, and JNK signaling pathways, but not the p38MAPK kinase pathway which subsequently leads to NRF2 nuclear translocation and ultimately increases the expression of phase II enzymes (*Huang et al., 2011*). In contrast, Kaempferol upregulates LPS-mediated expression of HO-1 in human alveolar epithelial cells *via* the p38MAPK-NRF2 pathway and protects lung tissue from inflammatory injury (*Yang et al., 2022*) (Fig. 4).

With the exception of above-mentioned chemicals such as Panaxydol and Obacunone, which have been shown to modulate NRF2 to ameliorate ferroptosis, there are a large number of natural drugs that target NRF2 and its upstream signaling to counteract oxidative stress and alleviate inflammatory responses. For example, Sinomenine and

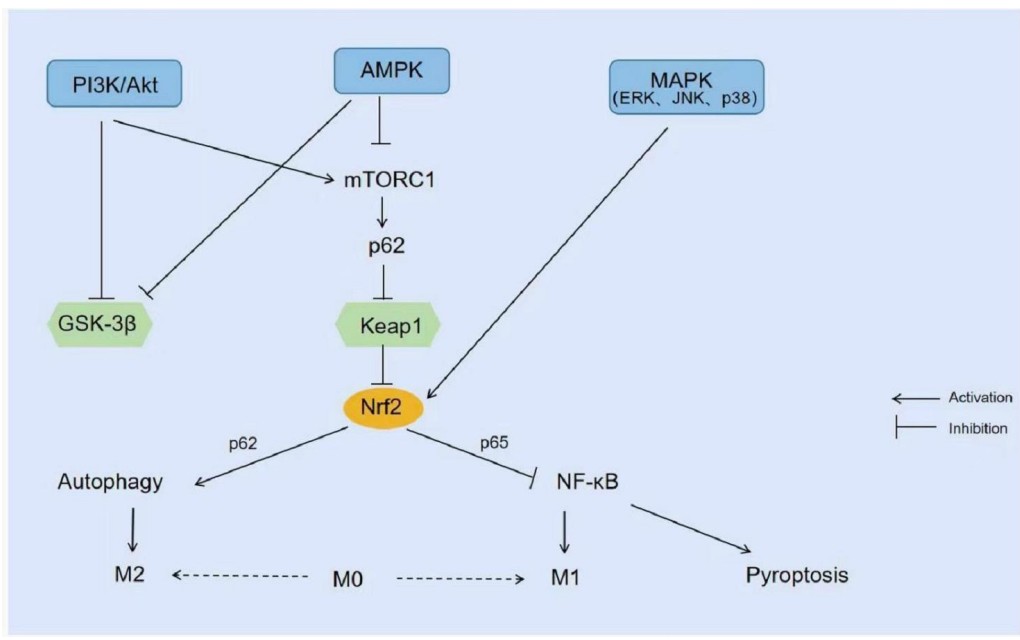

**Figure 4   Signal pathway related to NRF2.** PI3K/Akt inhibits GSK-3β or promotes the expression of mTORC1 and reduces the degradation of NRF2. AMPK also promotes the expression of NRF2 by inhibiting GSK-3β. MAPK (ERK1/2, JNK, and p38MAPK) activates NRF2 and promotes its nuclear transfer.

Cordycepin pretreatment increased the expression of NRF2, HO-1 and NQO1 in LPS-induced RAW264.7 cells (*Qing et al., 2018*; *Wang et al., 2020*). In contrast, Oridonin was able to regulate the phosphorylation of NRF2 upstream kinases (including Akt, JNK, p38MAPK, and ERK) which could promote NRF2 cytoplasmic accumulation and nuclear translocation, thereby attenuating LPS-induced ROS production in LPS-induced RAW 264.7 cells and inhibiting LPS-induced murine myeloperoxidase/ malondialdehyde (MPO/MDA) production, glutathione/superoxide dismutase (GSH/SOD) depletion and lung injury in mice (*Yang et al., 2019*). Similarly, Resveratrol activated the PI3K/Akt/NRF2 signaling pathway, induced HO-1 expression, and inhibited the expression of interleukin-18 (IL-18), MDA, and cysteine asparagin-3 in the lung tissues of rats in the CLP group, resulting in anti-inflammatory, anti-oxidative stress, and anti-apoptotic effects, as well as attenuating lung injury (*Wang et al., 2018*). There are also some acids, such as neochlorogenic acid from Morus alba leaf extract, triterpenic acid from loquat leaf extract, and ferulic acid from Ligusticum chuanxiong extract, which significantly inhibit the production of inflammatory mediators (including TNF-α, IL-1β, IL-6, and NO) and promote the expression of antioxidant enzymes (including SOD and HO-1) through activating the AMPK/NRF2 pathway, showing effective anti-inflammatory and antioxidant effects (*Gao et al., 2020*; *Jian et al., 2020*; *Wu et al., 2021*). Thus, the in-depth study of these natural drugs is also an important direction to combat ALI from the perspective of combating oxidative stress and improving ferroptosis.

There is usually a complex and intertwined network of mechanisms involved in disease development. It is not difficult to find that NRF2 is the intersection of multiple programmed cell deaths (ferroptosis, autophagy and pyroptosis) in ALI and is also a common regulatory point downstream of multiple signaling pathways. Therefore, further research on NRF2 is expected to be a breakthrough in combating ALI/ARDS.

## Abbreviations

| | |
|---|---|
| **ALI** | acute lung injury |
| **ARDS** | acute respiratory distress syndrome |
| **I/R** | ischemia/reperfusion |
| **LPS** | lipopolysaccharide |
| **OA** | oleic acid |
| **RILI** | radiation-induced lung injury |
| **NRF2** | transcription factor NF-E2-related factor 2 |
| **AP-1** | activator protein 1 |
| **CNC-bZIP** | cap'n'collar/basic leucine zipper |
| **Neh** | NRF2-ECHhomology |
| **ARE** | antioxidant response element |
| **CHD 6** | chromatin-deconjugating enzyme DNA-binding protein 6 |
| **RXRα** | retinoid X receptor alpha |
| **GSH** | glutathione |
| **γGCS** | glutamate cysteine synthetase |
| **GSS** | glutathione synthetase |
| **GCL** | glutamate cysteine ligase |
| **GPX4** | glutathione peroxidase 4 |
| **GSR** | glutathione reductase |
| **GSSH** | oxidized glutathione |
| **SLC7A11** | solute carrier family 7 member 11 |
| **NADPH** | nicotinamide adenine dinucleotide phosphate |
| **G6PD** | phosphate dehydrogenase |
| **PGD** | phosphogluconate 6 dehydrogenase |
| **TALDO 1** | transaldolase |
| **MEl** | malic enzyme I |
| **PHGDH** | phosphoglycerate dehydrogenase |
| **IDHl** | isocitrate dehydrogenase I |
| **HO-1** | heme oxygenase 1 |
| **FPN** | iron pump protein |
| **FTH 1/FLH** | the light and heavy chains of ferritin |
| **HRD 1** | HMG-CoA reductase degradation 1 |
| **RBXl** | Cul 3 RING-box 1 |
| **ER** | endoplasmic reticulum |
| **HNE** | hydroxynonenal |
| **PUFAs** | polyunsaturated fatty acids |
| **PL** | phospholipid radicals |
| **PLOO** | phospholipid peroxyl radicals |

| | |
|---|---|
| **PLOOH** | phospholipid peroxide |
| **PLO** | alkoxyphospholipid radicals |
| **IRP1/2** | Iron regulatory protein 1/2 |
| **Prominin2** | ferritin transfer protein |
| **TF/TFR-1** | transferrin/transferrin receptor |
| **NCOA4** | Nuclear receptor coactivator 4 |
| **AECC** | American-European consensus definition |
| **ROS** | oxygen species |
| **Fer-1** | Ferrostatin-1 |
| **IL** | interleukin |
| **TNF** | tumor necrosis factor |
| **HIF-1α** | hypoxia inducible factor-1α |
| **ALOX** | lipoxygenase |
| **BAP1** | ubiquitin carboxyl-terminal hydrolase |
| **MLL4** | histone-lysine N-methyltransferase 2B |
| **SCD1** | stearoyl-coenzyme A desaturase 1 |
| **IFN-γ** | interferon gamma |
| **ACSL4** | acyl-CoA synthetase long-chain family member 4 |
| **SLE** | Systemic lupus erythematosus |
| **MDA** | malondialdehyde |
| **IIR** | Intestinal I/R |
| **IA** | isoliquiritin apioside |
| **LDHA** | Lactate dehydrogenase A |
| **PFKL** | phosphofructokinase L |
| **PGK1** | phosphoglycerate kinase 1 |
| **RILF** | radiation-induced lung fibrosis |
| **TGF** | transforming growth factor |
| **PTGS2** | prostaglandin endoperoxide synthase 2 |
| **Bcl-2** | B-cell lymphoma-2 |
| **NF-kB** | inhibited nuclear factor-κB |
| **NLRP3** | NOD-like receptor protein 3 |
| **PI3K/Akt** | phosphoinositide 3-kinase/Akt |
| **AMPK** | 5′-AMP-activated protein kinase |
| **MAPK** | mitogen-activated protein kinase |
| **GSK-3β** | glycogen synthase kinase 3β |
| **mTORC 1** | mammalian target of rapamycin 1 |
| **ERK1/2** | extracellular signal-regulated kinase1/2 |
| **JNK** | c-Jun N-terminal kinase |
| **MPO** | myeloperoxidase |
| **SOD** | superoxide dismutase |

### Funding

This work was supported by the National Natural Science Foundation of China (No. 81973840 and No. 81273748). The funders had no role in study design, data collection and analysis, decision to publish, or preparation of the manuscript.

### Grant Disclosures

The following grant information was disclosed by the authors:
The National Natural Science Foundation of China: 81973840, 81273748.

### Competing Interests

The authors declare there are no competing interests.

### Author Contributions

- JiaLi Deng conceived and designed the experiments, prepared figures and/or tables, authored or reviewed drafts of the article, and approved the final draft.
- Na Li conceived and designed the experiments, prepared figures and/or tables, and approved the final draft.
- Liyuan Hao performed the experiments, analyzed the data, authored or reviewed drafts of the article, and approved the final draft.
- Shenghao Li performed the experiments, authored or reviewed drafts of the article, and approved the final draft.
- Nie Aiyu conceived and designed the experiments, analyzed the data, prepared figures and/or tables, and approved the final draft.
- Junli Zhang performed the experiments, prepared figures and/or tables, and approved the final draft.
- XiaoYu Hu conceived and designed the experiments, authored or reviewed drafts of the article, and approved the final draft.

### Data Availability

  This is a literature review and did not utilize raw data or code.

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
