# Peer review of "Transcription factor NF-E2-related factor 2 plays a critical role in acute lung injury/acute respiratory distress syndrome (ALI/ARDS) by regulating ferroptosis"

_PeerJ, doi:10.7717/peerj.17692_

## Round 0.1 · original submission · Major Revisions

I think the comments of reviewer-1 are particularly pertinent here. Please make a clear and cogent rationale for the importance of THIS review, and contextualize it; i feel the negative elements of the angle you present should also be made cogently clear to the reader.

These changes require a significant element of re-writing, hence I have marked this as a major revision. I hope you will embrace this, as the reviewers and I recognise the potential value of this work.

Reviewer 1 ·

Basic reporting

This review article explores how NRF2 modulates ferroptosis to mitigate ALI/ARDS, highlighting its role in regulating redox homeostasis and anti-inflammatory responses. By reviewing relevant literature and dissecting NRF2-regulated signaling pathways related to ferroptosis, it provides insights into potential therapeutic interventions for ALI/ARDS, offering a valuable theoretical foundation for clinicians and researchers in respiratory disease.

Experimental design

no comment

Validity of the findings

Section 1 - NRF2

This section predominantly describes the features and functions of NRF2 without offering critical analysis or discussion. Adding critical insights, such as limitations of existing research, conflicting findings, or areas requiring further investigation, would enrich the content and demonstrate a deeper understanding of the topic.

Section 2 - Lipid Peroxidation and Ferroptosis

While this section delves into the mechanisms of lipid peroxidation and ferroptosis, it lacks contextualization within the broader field of cell biology or pathology. Providing context by discussing the significance of these processes in disease development or physiological functions would help readers grasp their relevance.

Section 3 - NRF2 Involved in ALI/ARDS

The text in this section tends to repeat phrases and concepts, which can make the manuscript redundant. For instance, phrases like "ameliorating ALI/ARDS induced by multiple causes" and "attenuate the inflammatory response" are repeated multiple times.

While the text provides a comprehensive overview of NRF2's involvement in different models of ALI/ARDS, it could benefit from more conciseness. Some paragraphs contain excessive detail, making it challenging for obtaining the key information efficiently. Streamlining the content by focusing on essential points would improve clarity.

This section primarily focuses on the positive effects of NRF2 activation in mitigating ALI/ARDS and ferroptosis. However, it would be beneficial to include a discussion of potential challenges associated with targeting NRF2 as a therapeutic approach. Addressing limitations would provide a more balanced perspective and acknowledge areas for future research.

Section 5 - The protective effect of compound molecules on ALI models by regulating the NRF2 signaling pathway and ferroptosis

While the text mentions various compounds, it could benefit from providing more specific details about their mechanisms of action and experimental evidence supporting their efficacy.

Additional comments

Section number should be corrected; there is no section 4.

Reviewer 2 ·

Basic reporting

Basic reporting: In the submitted manuscript “NRF2 plays a critical role in ALI/ARDS by regulating ferroptosis”, the authors reviewed how NRF2 plays a role in ALI/ARDS, which is the intra- and extra-pulmonary causative factor-induced lung injury, by regulating ferroptosis. The functional role of a key gene, NRF2, which is associated with lung injury were showed and discussed. The authors validated that NRF2 might be involved in the development of ALI/ARDS via ferroptosis.

The manuscript is clearly written and sound, however, the story created from literature view is not sufficiently strong to explain the patho-mechanism of ALI/ARDS consequence of ferroptosis caused by NRF2. In addition, the authors need further discussion for the evidence of which NRF2 are directly and specifically involved in the ferroptosis for the development of ALI/ARDS

Monor comments
1) L68: Yuei Wai Kan et al. would be Moi P et al.
2) L74: samll is not correct.
3) L77: Neh 3 should be Neh3.
4) L86 and L91: Neh 4 and Neh 5 should be Neh4 and Neh5.
5) L90: Neh 7 should be Neh7.
6) L335: 5 should be 4.
7) L365 and L370: et al. would be etc.
8) L416: GSK-3btoo. should be GSK-3b too.

Experimental design

Study design is clear and well organized in the relation between NRF2 and ALI/ARDS, and ferroptosis and ALI/ARDS. However, the association between NRF2 and ferroptosis should also be analyzed to validate the function of NRF2 as the ferroptosis-associated molecule and the role in the development of ALI/ARDS.

Validity of the findings

As I mentioned above, if there is a weakness, the authors should clarify whether NRF2 is a ferroptosis-associated specific protein in ALI/ARDS.

---

## Round 0.2 · Minor Revisions

Thanks for the revisions, which are largely clear and acceptable. However, I do feel the comment of reviewer-2 has not been fully addressed:

'The manuscript is clearly written and sound, however, the story created from literature view is not sufficiently strong to explain the patho-mechanism of ALI/ARDS consequence of ferroptosis caused by NRF2. In addition, the authors need further discussion for the evidence of which NRF2 are directly and specifically involved in the ferroptosis for the development of ALI/ARDS

To suggest this will be the object of a further study is perhaps insufficient. Some attempt at addressing this issue needs to be made. This may take the form of a 'signpost' for the field.

I look forward to seeing this minor amendment.

---

## Round 0.3 · accepted · Accept

Thank you for tidying up the remaining issue.